# Mathematical Modeling of Oncolytic Virus Therapy Reveals Role of the Immune Response

**DOI:** 10.3390/v15091812

**Published:** 2023-08-25

**Authors:** Ela Guo, Hana M. Dobrovolny

**Affiliations:** Department of Physics & Astronomy, Texas Christian University, Fort Worth, TX 76109, USA

**Keywords:** mathematical model, interferon, cancer, parameter estimation, immune response

## Abstract

Oncolytic adenoviruses (OAds) present a promising path for cancer treatment due to their selectivity in infecting and lysing tumor cells and their ability to stimulate the immune response. In this study, we use an ordinary differential equation (ODE) model of tumor growth inhibited by oncolytic virus activity to parameterize previous research on the effect of genetically re-engineered OAds in A549 lung cancer tumors in murine models. We find that the data are best fit by a model that accounts for an immune response, and that the immune response provides a mechanism for elimination of the tumor. We also find that parameter estimates for the most effective OAds share characteristics, most notably a high infection rate and low viral clearance rate, that might be potential reasons for these viruses’ efficacy in delaying tumor growth. Further studies observing E1A and P19 recombined viruses in different tumor environments may further illuminate the extent of the effects of these genetic modifications.

## 1. Introduction

Cancer remains one of the leading causes of death in the US, claiming nearly 600,000 lives each year [1]. Globally, cancer accounts for almost 16% of all deaths [2]. Current treatments for cancer, such as chemotherapy, radiation therapy, and surgery, have significant physical and mental effects on the patient [3]. Due to these treatments’ low specificity, that is, they damage the healthy parts of the body in addition to the cancer, they introduce painful side effects during the treatment process [4]. For this reason, researchers have been investigating alternative treatments for cancer.

Oncolytic viruses (OVs) have emerged as a potential cancer treatment option [5], with the advantages of high specificity and few adverse effects [6]. Several Phase I and II clinical trials using OVs on a variety of different cancers have been conducted in recent years. Several trials have shown promising success of various strains of OVs in eradicating tumors [7]. China approved the use of adenovirus H101, known under the name Oncorine, for treatment of head and neck cancers in 2005 [8]. About a decade later in the USA, the FDA approved T-VEC, a modified herpes virus, for the treatment of inoperable melanomas [9]. Current OVs struggle to perform well in in vivo environments. Systemically introduced OVs tend to be eliminated by the host’s body before they reach the tumor, and when introduced locally to the tumor site, the viruses struggle to permeate fully throughout the tumor matrix [10]. When the OVs do manage to infect and kill tumor cells, the stimulated immune response, in addition to anti-tumoral responses, also produces an anti-viral response. This results not only in cancer cell death, but also removal of the virus before the tumor is fully infected [11]. Both aforementioned issues can result in incomplete remission of the tumor, allowing it to grow back [12].

Oncolytic adenoviruses (OAds), a subtype of oncolytic viruses, have been studied extensively due to their abilities to replicate rapidly and induce an immune response [5,13,14,15]. Current research shows that OAds are capable of reducing tumors, but cannot completely eradicate them, allowing the tumor to grow back after a period of partial remission [16,17]. Various genetic modifications intended to raise the efficacy of OAds and solve some of the OVs’ shortcomings have been researched, including ways to diffuse the virus more thoroughly within a solid tumor [18] or enhancing viral replication [19]. Complex tumor stroma can inhibit the spread of the virus, but changes to the E1A gene have been shown to overcome this issue [20,21]. The addition of the RNA inhibitor suppressor gene P19 to the viral genome has been shown to enhance viral replication in insect cells [21], and in adenoviruses specifically, significantly amplified adenovirus replication and oncolytic activity [22].

The current understanding of the complex dynamics determining the effectiveness of OVs is limited, but mathematical models can help illuminate and quantify the underlying factors in oncolytic virus kinetics. Mathematical models for oncolytic viruses have been adapted from a set of ordinary differential equations used to describe virus dynamics, with varying levels of complexity [23,24]. Simpler models [23] assume perfect integration of the virus with tumor cells, often unachievable with solid tumors, while more complex models consider the effects of antiviral immune responses and anti-tumor responses triggered by the virus [24,25,26,27]. Various studies have been performed using these models to quantify the parameters involved in oncolytic virus kinetics, comparing parameter values for different treatment outcomes [28,29].

Some studies have considered the effect of immune response on the tumor–virus system, developing mathematical models that account for adaptive and delayed immune responses [25,26,30,31,32,33,34]. The immune response hinders viral replication, but also can have an anti-tumoral effect, leading to different outcomes based on the relative strength of these effects [26,32]. Depending on the exact mathematical implementation of these effects, the region of tumor eradication can be small [30,33,35] or can show immune response-driven multi-stability [34]. Models that have been fit to clinical data suggest that the innate immune response can severely diminish OV’s capacity to control a tumor [26,31].

In this paper, we continue building on this topic, comparing differences in parameters between different OAd treatments in live murine models. We fit mathematical models of OV dynamics to previously published data from OAd-treated xenograft tumors and find that the data are best explained by a mathematical model that includes an immune response.

## 2. Materials and Methods

### 2.1. Experimental Data

Tumor data were obtained from the paper *Novel group C oncolytic adenoviruses carrying a microRNA inhibitor demonstrate enhanced oncolytic activity in vitro and in vivo* by Doerner et al. [16], which observed the effects of novel OAds in mice with xenografted A549 lung tumors. In this study, recombineered P19 expressing adenoviruses based on adenovirus types 1, 2, and 6 were injected into different groups of mice. These three re-engineered viruses also had higher tumor selectivity and permeability due to a modification in the E1A gene [16]. A group of mice was injected with the HAd5 virus, and another was injected with the H101 virus, an OAd treatment for head and neck cancers approved in China [36]. Each treatment group initially consisted of five mice, which were sacrificed once the tumor ulcerated or reached a mass of 1000 ${\mathrm{mm}}^{3}$. As such, the data do not represent the progression of tumor growth in a single animal, they are an average of the five volumes measured from each mouse. Measurements of the tumors were taken using calipers for up to 160 days, and the volumes were calculated based on the tumor’s diameters. A more detailed explanation of methods and data collection is available in [16]. We use the data from Figure 6A, which contains the average cancer volume data in the live mouse models. Data were extracted using WebPlotDigitizer (https://automeris.io/WebPlotDigitizer/ (accessed on 16 June 2023)).

### 2.2. Model Fitting and Statistical Analysis

We evaluate two different mathematical models in fitting the treatment data to see if the added complexity of accounting for an immune response is justified. We initially investigate a basic model of OV therapy without any immune response,
(1)$$\begin{array}{ccc} \frac{dT}{dt} & = & \lambda T-\beta TV\\ \frac{dE}{dt} & = & \beta TV-kE\\ \frac{dI}{dt} & = & kE-\delta I\\ \frac{dV}{dt} & = & pI-cV. \end{array}$$

The tumor (*T*) grows at a rate λ. The virus (*V*) infects the tumor at rate β. The infected cells then enter the eclipse phase (*E*), and become actively infectious after a time $\frac{1}{k}$. Infected tumor cells (*I*) die at a rate δ and produce more viral particles at rate *p*. The virus is cleared from the cells at rate *c*.

We also use a model that includes an interferon response,
(2)$$\begin{array}{ccc} \frac{dT}{dt} & = & \lambda T-\beta TV\\ \frac{dE}{dt} & = & \beta TV-kE\\ \frac{dI}{dt} & = & kE-\delta I\\ \frac{dV}{dt} & = & I\left(\frac{p}{1+\epsilon F}\right)-cV\\ \frac{dF}{dt} & = & V-\alpha F. \end{array}$$

Here, the virus stimulates an interferon response (*F*), where ϵ characterizes the virus’ sensitivity to interferon, and α is the decay rate of interferon. We first investigate the addition of an interferon response since the mice used in the original study are CB17 mice that have a naturally occurring mutation that causes severe combined immunodeficiency (SCID). These mice are not capable of mounting a significant T cell or B cell response, so the adaptive immune response should not play a role in the viral infection in these animals. We chose to model the effect of interferon as reducing viral production, as has been done in other modeling studies [37,38,39]; although other assumptions for the effect of interferon are also possible [40,41]. This model has two more parameters than the basic model.

In order to estimate the model parameters, we fit both these models. Model fitting is a standard procedure whereby parameter values are systematically varied in an attempt to match the model curve to the experimental data. We quantitatively assess how well the model matches the data by calculating the sum of squared residuals (SSR) between the model and observed values,
(3)$$\mathrm{SSR}=\sum_{i=1}^{n}{\left(y_{i}-f\left(t_{i}\right)\right)}^{2},$$
where *n* is the number of data points, $y_{i}$ is the experimentally observed tumor volume at time *t*, and f(t) is the model’s predicted tumor mass at time *t*. The parameter values that lead to the minimal value of SSR are considered the best fit parameters estimates.

Since not all viral particles from the initial injection of virus manage to infect cells, we consider $V_{0}$, or the initial amount of virus in the system, as a free parameter. Because infection begins when the virus is first introduced to the tumor, we assume the initial volume of eclipse phase cells E(0), actively infectious cells I(0), and amount of interferon F(0) are 0. Fitting was conducted in two stages with the control tumor used to estimate the base growth rate. Treated tumors were then assumed to have that same growth rate and other viral parameters were estimated using fits to treated data. The list of parameters that were estimated using this fitting procedure and the biological meaning of the parameters is included in Table 1.

For the fitting of the tumor kinetics model, we use various aspects of the scipy Python package. First, the ordinary differential model of the tumor kinetics is solved by the odeint function of the scipy.integrate sub-package, and the minimize function of the scipy.optimize sub-package minimizes the sum of squared residuals (SSR) of the model to the data. We used the Nelder–Mead minimization function to identify the best-fit parameters [42]. Plots were generated using the matplotlib Python package [43].

To determine the 95% confidence intervals for the best-fit parameters and SSR’s, we use 1000 bootstrap replicates [44]. Bootstrapping was also used to determine possible parameter distributions. Distribution graphs were made using pyplot from the matplotlib Python package [43].

## 3. Results

### 3.1. Control Tumor

We used an exponential model to fit the untreated tumor curve. The fit of the model is shown in Figure 1 along with the exponential growth rate estimate. We find that untreated A549 tumors have a growth rate of 0.0996/day (0.0960–0.100/day 95% CI). The base tumor growth rate will be fixed to this value for subsequent fits to the treated data.

### 3.2. Model without an Immune Response

The best-fit curves for the model without an immune response are displayed in Figure 2, and best fit parameters with 95% confidence intervals are displayed in Table 2. Parameter correlation plots and histograms are included in the Appendix A. While the model is capable of capturing the plateaus in tumor growth caused by the virus, along with the eventual rapid growth once the virus fails, it is not capable of capturing the smaller growth and shrinkage in the tumor seen in the early part of the time series. For this reason, we explored a model that included an immune response to see if it might be able to capture the finer structure of the tumor time course.

### 3.3. Model with an Immune Response

The best-fit model curves for the model including an immune response are displayed in Figure 3, and best fit parameters with 95% confidence intervals are displayed in Table 3. Parameter correlation plots and parameter histograms are included in the Appendix A. The model that includes an immune response is able to capture the small changes in growth of the tumor early in the time course and we see a substantial reduction in the SSR for fits with this model compared to the basic model fits.

Many of these parameters have not been previously estimated for OAds. The viral clearance rate ranges from ∼0.05–0.38/day. The infected cell lifespan ranges from ∼2–14 days. A previous study measured adenovirus-infected cell lifespans of ∼1–4.5 days [45], which is generally shorter than the lifespans measured here. We note, however, that the rate of transfer from the eclipse state and the infected cell death rate are very similar for all five strains, suggesting that it is difficult to distinguish the values of these two parameters. We also find that decay of interferon is very slow, with α ranging from 4.8 × $10^{-13}$–2.4 × $10^{-7}$/day. The most effective OAds (Ad6d24.P19, Ad1d24.P19, Ad2d24.P19) in delaying tumor progression share a few characteristics relative to the other viruses: a higher infection rate (β), a high value of (ϵ), and a lower viral clearance rate *c*.

### 3.4. Comparison of Models

We use the Akaike Information Criterion to evaluate which model fits the data better. The AIC assigns a numerical value to each model based on the following equation,
(4)$$\begin{array}{ccc} AIC & = & n\ln\left(\frac{\mathrm{SSR}}{n}\right)+2k, \end{array}$$
where *n* is the number of data points for each set of tumor data, SSR is the sum of squared residuals from the best fit model, and *k* is the number of parameters [46]. Having more parameters penalizes the model, with lower AIC indicating a better-fitting model [24,47]. Models with an approximate difference of two units in their AIC scores are not considered substantially different.

AIC values for the two models for each virus are given in Table 4. The lowest AICs for each model have been bolded. With the exception of the H101 adenovirus, the AIC for the model with immune response is lower than that of the basic model, indicating that the immune model provides a better fit. This also indicates that the improvement in the model fit justifies the extra complexity of the model from the addition of the immune response.

### 3.5. Comparison of Parameter Estimates for Different OAds

Figure 4 displays the logs of the bootstrapped parameter values, grouped by parameter for the different viruses. H101 is a clear outlier in its values of infection rate (β) and production rate (*p*). This virus is also the least sensitive to the effect of interferon (lowest ϵ). Among the remaining viruses, Ad1d24.P19 has particularly low infected cell death rates and slow viral clearance. This is the virus that showed the longest delay before resumption of tumor growth—long-lived infected cells and slow viral clearance could account for that.

Statistical tests were performed using the Mann-Whitney U-test by taking random samples of ten data points from the 1000 bootstrap replicates, testing with an alternative hypothesis that the median values are not equal, and recording the *p*-values. The test was repeated 1000 times for each comparison, and the average *p*-value of the 1000 trials is listed in the tables in the Appendix A. Significantly low *p*-values have been bolded. The results here reflect the histogram representation, as viruses with high overlap in parameter values tended to share higher *p*-values (i.e., their parameter values were very likely to be similar). Significant *p*-values are any values below a value of 0.05. A short summary of major differences is listed below.

The H101 OAd showed the most differences in parameter values from the other viruses. H101 had significantly different β, *p*, and ϵ values from all the other viruses. The viral clearance rate, *c*, of H101 was significantly higher than all viruses except Ad5.P19, while the interferon clearance rate was significantly different from all viruses except Ad6d24.P19. All of these differences likely contribute to H101’s poor performance in delaying tumor growth as compared to the other viruses.

Several of the parameters appeared to be fairly distinct for all of the viral strains examined here. The viral production rate, *p*, was significantly different for all viral strains. The sensitivity of virus to interferon. ϵ, was significantly different for all strains except Ad5.P19 and Ad1d24.P19, which show a substantial amount of overlap. Finally, the viral clearance rate was significantly different for all strains except Ad5.P19/H101 and Ad2d24.P19/Ad6d24.P19.

There were also two parameters that showed a few significant differences. The infected cell death rate, δ for Ad1d24.P19 was significantly different from the death rate of other viral strains, which all had similar death rates. The experiments of Doerner et al. support this idea, showing that Ad1d24.P19 has little effect on cell viability at 5 and 7 dpi in many cell lines [16]. Ad1d24.P19 also had a significantly different transition rate from the eclipse phase, *k*, than all other strains except Ad2d24.P19. These factors likely explain the particularly long plateau in tumor growth for this strain of virus. The virus is infecting the tumor cells, which is slowing down proliferation of the cancer cells, but the infected cells die slowly, so a balance is reached for a period of time where, overall, the tumor does not grow.

### 3.6. Role of the Immune Response in Suppressing Tumors

While the experimental data suggest that none of the viruses eliminate the tumor, mathematical models allow us to extrapolate beyond the end of the experimental data. Figure 5 shows the total tumor volume (sum of uninfected, eclipse, and infected cells) predicted by the two models up to 200 days post-infection. For all viruses, the model without an immune response predicts that the tumor will continue to grow and viral treatment fails to eliminate the tumor. The viral time courses for this model suggest that the viral titer only decays from its initial value, so that an infection never really takes hold in the tumor. The model with an interferon response, however, suggests that the initial small bump observed in the experimental data is a pre-cursor to a larger oscillation that can dramatically drive down the size of the tumor. This is a seemingly counter-intuitive result since the interferon response in this model is assumed to have only an antiviral effect and does not directly impact the tumor. However, the interferon response in this case slows, but does not completely stop, viral production allowing for long lasting viral infections that can infect and kill more cancer cells.

## 4. Discussion

We find that an oncolytic virus model without an immune response is unable to reproduce experimentally observed tumor progression. The model without an immune response fails to capture the small oscillation present in the treated tumor data and also predicts that virus will simply decay. Other studies have also found it necessary to incorporate an immune response to accurately capture the behavior of oncolytic viruses, although the mathematical implementation of the immune response differed from the implementation presented in this manuscript [31,48]. The importance of the immune response is also supported by experimental studies that have found that due to the interaction of OVs with the tumor system, an immune response will almost always be elicited [49,50]. In many cases, the interferon response has been observed to hinder the efficacy of OVs by inducing an antiviral effect [51,52,53,54]. However, many cancer cell lines have a reduced sensitivity to interferon [53,55,56,57]; therefore, an interferon response that slows viral replication without completely suppressing it, as predicted by our model, is possible.

We find that the most effective OAds in delaying tumor progression, namely serotypes -1, -2, and -6, share several characteristics not seen in the parameters of the other two viruses. Each effective OAd had an infection rate (β) with an order of magnitude of 1, a relatively high sensitivity to interferon (ϵ), and a slow viral clearance rate. This potentially indicates that the viruses had greater success in delaying tumor growth due to higher infection rates and less clearance of the virus, allowing it to proliferate further. Other studies have also noted the importance of high viral infectivity and low viral clearance in tumor suppression [48,58,59]. The high value of ϵ is consistent with the idea that the optimal immune response is one that builds slowly, allowing viral replication to slow, but not stop replication, helping to prolong the infection. This is further corroborated by the work of Mahasa et al., who note that recruitment of the body’s innate immune response might aid in suppressing tumor growth [25]. Complete and fast permeation throughout the tumor is critical since the host’s innate immune response initiates quickly, and can quickly suppress the OV if the viral infection is not sufficiently established and we are aiming for the immune response to not fully suppress the infection, but to slow down the initial quick infection.

However, the model we used only evaluates one aspect of the various immune responses in the body; therefore, more complex models that consider responses other than interferon response, and differentiate between antiviral and anti-tumor interferon responses, should be considered as well. Various studies that consider both antiviral and anti-tumor immune responses [25,60] indicate that the responses act in opposition, where the anti-tumor response helps with tumor suppression, while the antiviral response clears the OV, allowing the tumor to proliferate. Modeling this competition between the anti-tumor and antiviral innate immune responses will give a clearer picture of OV–tumor interaction. Other components of the immune response also play a role in OV’s ability to clear tumors and should be considered in modeling studies. For example, natural killer cells also have both antiviral and anti-tumor effects [61,62] that might enhance or mitigate the effect of interferon. A better understanding of the role of the immune response in enhancing or limiting OV effectiveness is becoming more crucial as researchers are starting to investigate combination therapy using OVs and immunotherapies [26,40,63].

Conclusions for virus effectiveness in humans are limited by the fact that the data were from animal models, and only observe efficacy in one type of tumor. Adenovirus replication in mouse models is known to be limited without inserting the appropriate cell surface receptors [64,65], particularly when trying to study OV therapy in immunocompetent mice. Even when OAds show good replication in the mouse model, we need to keep in mind the differences in immune responses between mice and humans [66,67], particularly when there seems to be a fine balance between too much of an immune response, which will halt replication of the virus, and not enough of an immune response, which will fail to establish a long-lasting viral infection capable of killing the tumor.

Despite the limitations, our model has identified virus–host characteristics that lead to improved OV performance and has highlighted the importance of incorporating an immune response. Further studies that observe how the P19 and E1A gene modifications perform in different tumor environments should be considered, but the RNAi suppression caused by P19 shows a promising path forward in eliminating antiviral response.

## Figures and Tables

**Figure 1 viruses-15-01812-f001:**
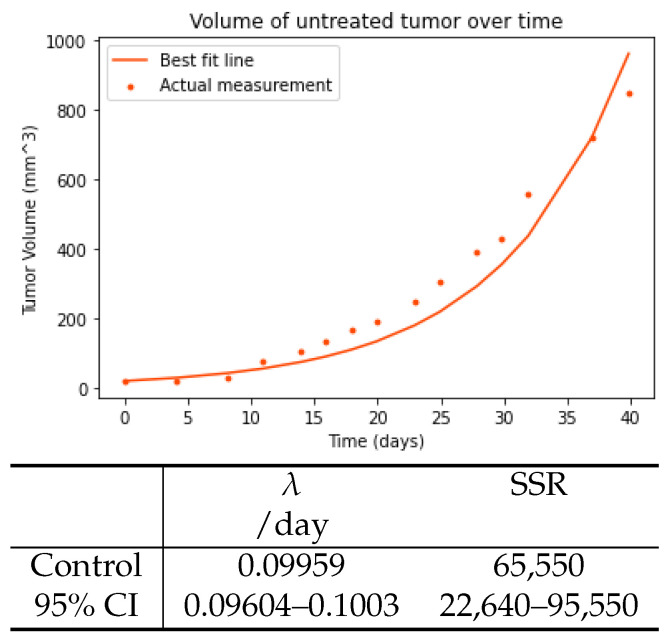
Exponential model fit and best fit parameters of the untreated curve from Figure 6A of [16].

**Figure 2 viruses-15-01812-f002:**
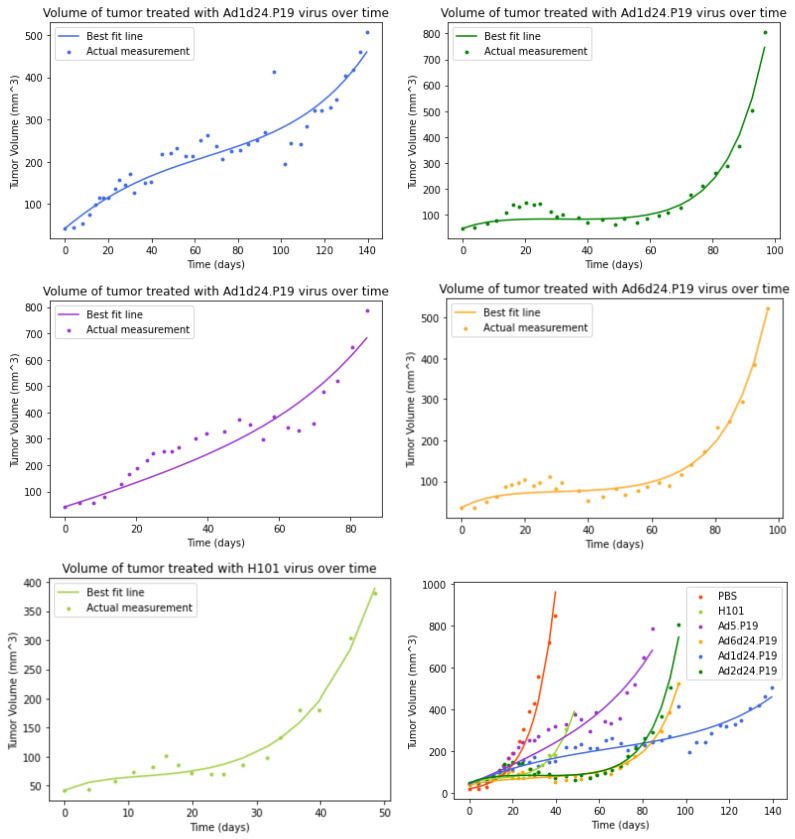
Graphs of tumor data and model without immune response. (**top left**) Ad1d24.P19 (**top right**) Ad2d24.P19 (**center left**) Ad5.P19 (**center right**) Ad6d24.P19 (**bottom left**) H101 (**bottom right**) all treatment curves plotted together.

**Figure 3 viruses-15-01812-f003:**
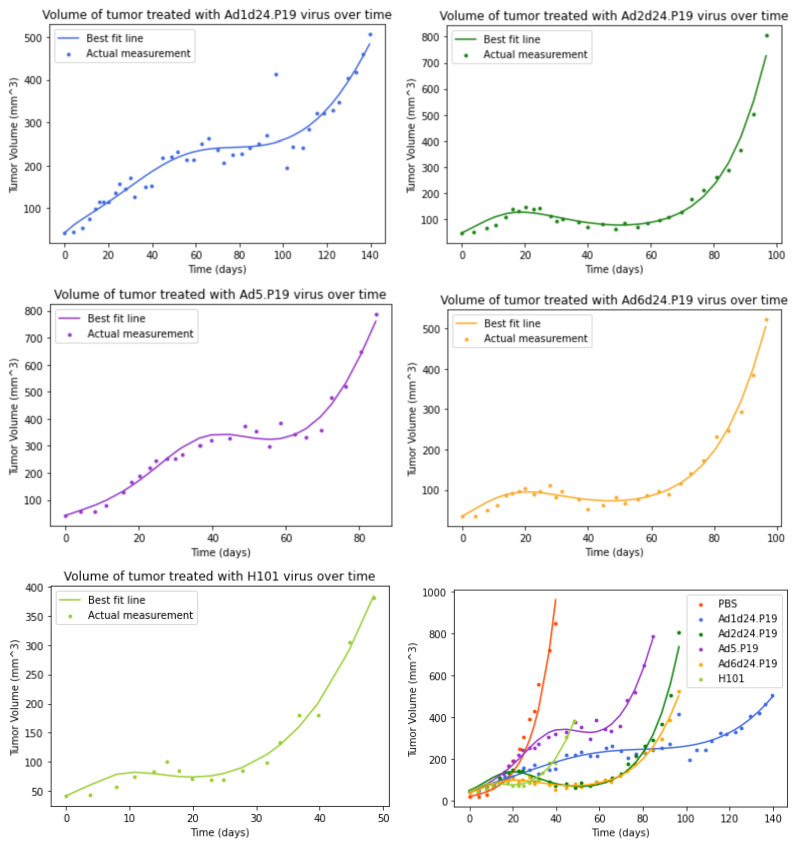
Graphs of tumor data and fits of the model with an immune response. (**top left**) Ad1d24.P19 (**top right**) Ad2d24.P19 (**center left**) Ad5.P19 (**center right**) Ad6d24.P19 (**bottom left**) H101 (**bottom right**) all treatment curves plotted together.

**Figure 4 viruses-15-01812-f004:**
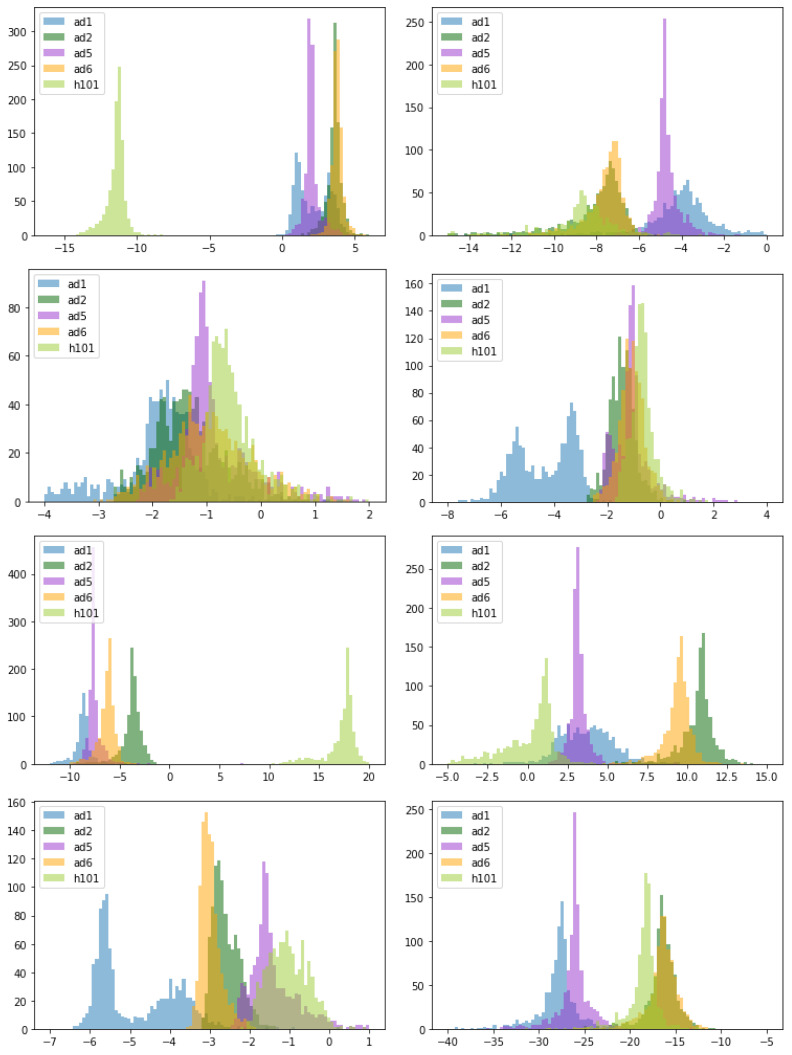
Histogram plots of logs of parameter distributions: (**top right**) β, (**top left**) V(0), (**second row left**) *k*, (**second row right**) δ, (**third row left**) *p*, (**third row right**) ϵ, (**bottom left**) *c*, and (**bottom right**) α.

**Figure 5 viruses-15-01812-f005:**
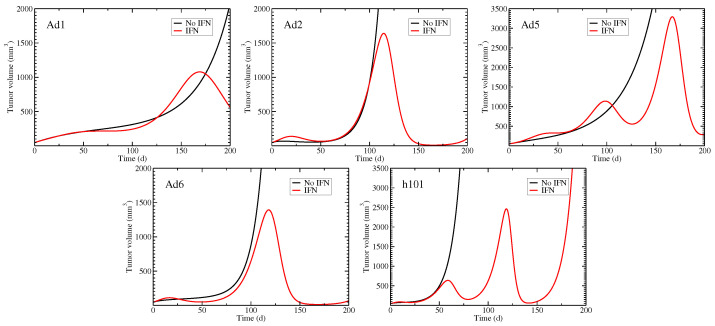
Predicted tumor volume for the models with and without interferon.

**Table 1 viruses-15-01812-t001:** Model parameters estimated by the fitting process.

Parameter	Meaning
β	Viral infection rate
V(0)	Initial amount of virus
*k*	Transition rate from eclipse to infectious cells
δ	Death rate of infectious cells
*p*	Viral production rate
ϵ	Sensitivity to interferon
*c*	Viral clearance rate
α	Interferon clearance rate

**Table 2 viruses-15-01812-t002:** Parameter estimates for model without an immune response.

Adenovirus	β	V(0)	k	δ	*p*	*c*	SSR
	**/day**		**/days**	**/days**	**/days·mm** ${}^{3}$	/**days**	
Ad1d24.P19	$7.005\times 10^{-3}$	15.68	0.02882	0.03211	$1.217\times 10^{-14}$	$2.598\times 10^{-3}$	53,730
95% CI	(5.402–9.487) $\times \;10^{-3}$	11.28–19.97	0.0–0.09937	0.0–0.2212	(0–1.345) $\times \;10^{-13}$	(3.923–49.29) $\times \;10^{-4}$	20,460–91,360
Ad2d24.P19	0.4754	0.4621	0.04066	0.4067	$2.089\times 10^{-13}$	0.02928	30,240
95% CI	0.2514–0.8510	0.2600–0.8649	0–2.325	0–1.494	(0–1.256) $\times \;10^{-11}$	0.01892–0.06247	15,100–37,250
Ad5.P19	0.2704	0.3151	0.04268	0.4268	$3.207\times 10^{-13}$	$1.824\times 10^{-3}$	105,000
95% CI	0.1215–0.5603	0.1463–0.6748	0.0–1.625	0.0–0.9863	(0.0–9.800) $\times \;10^{-12}$	(0.0–9.683) $\times \;10^{-3}$	50,200–133,200
Ad6d24.P19	286.3	$8.087\times 10^{-4}$	$1.975\times 10^{-3}$	200.0	$7.261\times 10^{-5}$	0.03025	12,530
95% CI	195.6–432.7	(55.35–1.201) $\times \;10^{-3}$	0–0.01935	(6.709–13,540,000) $\times \;10^{-4}$	(0–6.589) $\times \;10^{-4}$	0.02321–0.05481	6741–14,980
H101	107.1	$1.716\times 10^{-3}$	0.1321	0.1321	1.704 $\times 10^{-17}$	0.06521	3511
95% CI	50.25–270.5	(7.213–40.17) $\times \;10^{-4}$	0.01490–3.517	(4.935–2753) $\times \;10^{-3}$	(0–3.275) $\times \;10^{-16}$	0.03307–0.1552	1337–4795

**Table 3 viruses-15-01812-t003:** Parameter estimates for a model with an immune response.

Adenovirus	β	V(0)	k	δ	*p*	ϵ	*c*	α	SSR
	**/day**		**/day**	**/day**	**/days · mm** ${}^{3}$		**/day**	**/day**	
Ad1d24.P19	14.51	$8.793\times 10^{-3}$	0.07406	0.07406	$1.243\times 10^{-4}$	47.58	0.06857	$4.854\times 10^{-13}$	41,540
95% CI	1.691–76.10	0.0–0.2957	0.01–0.6070	(1.605–62.22) × $10^{-3}$	(0–4.427) $\times \;10^{-4}$	1.841–1042	(2.361–65.55) × $10^{-3}$	(0–1.637) × $10^{-11}$	19,420–120,700
Ad2d24.P19	39.00	$6.354\times 10^{-4}$	0.2346	0.2333	0.02650	58,910	0.07022	$5.804^{-8}$	17,790
95% CI	10.93–100.9	(1.459–18,480) × $10^{-7}$	0.08211–1.161	0.08950–0.6671	(9.445–1108) × $10^{-4}$	2480–246,700	0.04706–0.1565	(5.693–131,800) × $10^{-11}$	6077–26,180
Ad5.P19	7.297	$8.440\times 10^{-3}$	0.3198	0.3198	$4.659\times 10^{-4}$	22.87	0.2127	$5.053\times 10^{-12}$	15,000
95% CI	2.833–34.15	0.002776–0.03845	0.1335–3.397	0.1211–2.588	(1.917–24.92) × $10^{-4}$	7.208–84.24	0.1046–1.321	(6.966–142,700) × $10^{-15}$	6554–18,850
Ad6d24.P19	45.35	$7.078\times 10^{-4}$	0.3189	0.3162	$2.073\times 10^{-3}$	6295	0.04982	$2.421\times 10^{-7}$	4640
95% CI	22.46–114.2	(3.005–169.6) × $10^{-5}$	0.08875–1.923	0.1195–0.9148	(1.605–64.76) × $10^{-4}$	679.2–21,930	0.03503–0.1028	(7.127–691,000) × $10^{-11}$	2041–6123
H101	$1.223\times 10^{-5}$	$1.460\times 10^{-4}$	0.4649	0.4647	$6.166\times 10^{7}$	2.884	0.3752	$8.022\times 10^{-9}$	2826
95% CI	(1.939–28.44) × $10^{-6}$	0–$1.457\times 10^{-3}$	0.2496–2.115	0.2428–1.390	(4.176–2043) × $10^{5}$	0.02134–10.85	0.1542–0.9240	0–1.335 × $10^{-7}$	978.5–3644

**Table 4 viruses-15-01812-t004:** Comparison of AIC values for models with and without immune response.

Adenovirus	With Immune Response	Without Immune Response
Ad1d24.P19	**305.7**	312.5
Ad2d24.P19	**208.1**	219.5
Ad5.P19	**181.3**	227.9
Ad6d24.P19	**167.2**	193.0
H101	**102.9**	**102.6**

## Data Availability

Data and computer code are available at https://github.com/hdobrovo/Oncolytic_immune (accessed on 16 June 2023).

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
