# Peer review of "Mathematical Modeling of Oncolytic Virus Therapy Reveals Role of the Immune Response"

_viruses, 2023, doi:10.3390/v15091812_

Round 1
Reviewer 1 Report
In this manuscript, Ela Guo and Hana M. Dobrovolny worked out a differential equation (ODE) model of tumor growth inhibited by oncolytic virus activity. The use the data from a previously published research article (Doerner et al.,2022) on the effect of genetically engineered OAds in A549 lung cancer tumors in murine models. The authors concluded that their model “has identified virus-host characteristics that lead to improved OV performance” and “has highlighted the importance of incorporating an immune response” in their predictive model.
OVs are nowadays considered a type of immunotherapy. Infected cancer cells have shown to activate antiviral (I) and cell death pathways (II), contributing to the release of type-I interferons, cytokines and chemokines as well as DAMPS and PAPMs. Altogether, the direct effect of OV over cancer cells leads to the remodeling of the immunosuppressive tumor microenvironment, facilitating the recruitment and engagement of the innate and adaptive immune cells towards the elicitation of an anti-tumor response.
In silico tools able to integrate preclinical and clinical data to predict whether a particular viral platform would be able to achieve a therapeutic outcome in a particular cancer are in need and will be essential to further propel the expansion of virotherapy of cancer.
To address Comments:
The referenced paper and source of data in this work (Doerner et al.,2022) use immunodeficient NOD.CB17-Prkdcscid/J (CB17) mice to assess oncolytic potential of different engineered OAds on an A549 lung carcinoma xerograph tumor model.
These animals are limited on their capacity to build up adaptive immune anti-tumor and antiviral responses. The authors noticed that “the model used only evaluates one aspect of the various immune responses in the body, so more complex models that consider responses other than interferon response, and differentiate between antiviral and anti-tumor interferon responses, should be considered as well”:
In this regard, despite acknowledging the caveats of the model, the claims made by the authors could be and should be further contextualized, using and referring to the information included in the original work -Doerner et al.,2022- to back up their conclusions:
- There are 5 different recombinant viruses with different biology -replication capacity, sensitivity to interferons….- . how does the biology of the viruses relate to the results inferred by the authors ODE model?
- In the original article the authors concluded: “Here we generated P19-expressing, selectively replicating (oncolytic) adenoviruses (…) . We show that Ad types 1-, 2-, and 6-based viruses display higher cell lysis capacity in several tumor cell lines of different origins. Furthermore, the expression of P19 resulted in up to 10-fold higher oncolytic capacity in cell culture assays compared with control viruses lacking P19 in cell culture assays. Expression of P19 translated into significantly higher replication rates in lung cancer cells. Finally, treatment with P19 expressing oncolytic Ad based on types 1, 2, and 6 resulted in significantly reduced tumor growth and improved survival compared with existing oncolytic virus H101”.
o Please elaborate if so how the ODE model is integrating this information and the data sources used from the original manuscript.
o Authors claim that the data used in this manuscript to develop their ODE comes from Figure 6. Please extend and elaborate how the parameters related to sensitivity to interferon, rate of infection and decay rate have been estimated.
Reviewer 2 Report
Comments for the Authors
viruses-2524274
Title: Mathematical Modeling of Oncolytic Virus Therapy Reveals Role of the Immune Response
Authors: Ela Guo and Hana M. Dobrovolny
Guo and Dobrovolny performed clinical data simulation using two models (with- and without interferon response) constructed based on an ordinary differential equation to compute the progression of tumor growth in conjunction with oncolytic virus activity. The clinical dataset was retrieved from the study (Doerner et al. 2022) conducted in mice with xenografted A549 lung tumors. Mice were infected with either re-engineered adenoviruses or HAd5 and H101viruses approved for treating other types of tumors. Collectively, Guo and Dobrovolny observed that the model coupled with the immune response fit the clinical dataset and claimed that (1) infection rate (β) with an order of magnitude of 1, (2) a relatively high sensitivity to interferon (e), and (3) a slow viral clearance rate were critical characteristics of effective oncolytic adenoviruses.
This manuscript is well written with a clear explanation of the rationale and the theme of this study.
I have only one comment concerning this manuscript: Apart from H101, Ad1d24.P19 appeared in a different pattern upon simulation shown in Figures 2, 3, and 4 compared to Ad2d24.P19 and Ad5.P19 and Ad6d24.P19. Can the authors discuss the cause of this difference? Is it due to the distinct genetic background of this chimeric virus from other viruses or other additional factors required to be taken into account to reinforce the mathematical models proposed in this work.
Reviewer 3 Report
In the manuscript by Guo and Dobrovolny, the authors present an ODE model of tumor growth inhibited by oncolytic virus activity to parameterize previous research on the effect of genetically re-engineered oncolytic adenoviruses in A549 lung cancer tumors in murine models. It is found that the data is best fit by a model that accounts for the immune response, the immune response providing a mechanism for the elimination of the tumor. It is also found that parameter estimates for the most effective oncolytic adenoviruses share characteristics that might be potential reasons for these viruses' efficacy in delaying tumor growth. Further studies are suggested to shed light on the extent of the effects of genetic modifications that are discussed in the paper.
The manuscript reads well, it is interesting and highly comprehensive. It suits well the topic of the special issue.
Minor comments:
1) The abbreviation PBS should be clarified to the reader.
2) A table of abbreviations that are used in the manuscript can make it easier for readers.
3) In lines 228 and 246: I suggest "so" -> "therefore", for authors discretion.
